# Isolation of Efficient Metal-Binding Bacteria from Boreal Peat Soils and Development of Microbial Biosorbents for Improved Nickel Scavenging

**Hanna Virpiranta [1],\*** , **Michal Banasik [2]** , **Sanna Taskila [1]** , **Tiina Leiviskä [1]** , **Maiju Halttu [1]** , **Ville-Hermanni Sotaniemi [1] and Juha Tanskanen [1]**

[1] Chemical Process Engineering, University of Oulu, PO Box 4300, 90014 Oulu, Finland; sanna.taskila@macon.fi (S.T.); tiina.leiviska@oulu.fi (T.L.); maijukhalttu@gmail.com (M.H.); ville-hermanni.sotaniemi@oulu.fi (V.-H.S.); juha.tanskanen@oulu.fi (J.T.)

[2] Department of Molecular Biotechnology and Microbiology, Gdańsk University of Technology, ul. Narutowicza 11/12, 80-233 Gdańsk, Poland; micbanasik@gmail.com

\* Correspondence: hanna.virpiranta@oulu.fi

**Abstract:** Boreal peatlands with low iron availability are a potential, but rarely studied, source for the isolation of bacteria for applications in metal sorption. The present research focused on the isolation and identification of Actinobacteria from northern Finland, which can produce siderophores for metal capture. The 16S rDNA analysis showed that isolated strains belonged to Firmicutes (*Bacillus* sp.) and Actinobacteria (*Microbacterium* sp.). The culture most efficiently producing siderophores in the widest array of the media was identified as *Microbacterium* sp. The most appropriate media for siderophore production by the *Microbacterium* strain were those prepared with glucose supplemented with asparagine or glutamic acid, and those prepared with glycerol or fructose supplemented with glutamic acid. The microorganism obtained and its siderophores were used to develop Sphagnum moss-based hybrid biosorbents. It was showed that the hybrid sorbent could bind nickel ions and that the nickel removal was enhanced by the presence of siderophores. Bacterial cells did not have a significant effect on sorption efficiency compared to the use of siderophores alone. The microbial biosorbent could be applied in the final effluent treatment stage for wastewater with low metal concentrations.

**Keywords:** metal-binding bacteria; siderophores; soil bacteria; Actinobacteria; *Microbacterium* sp.; biosorption; nickel

## 1. Introduction

Siderophores are compounds of low molecular weight (500–1500 Da) that exhibit extremely high affinity toward iron ($K_f > 10^{30}$). Depending on the moieties responsible for the binding of ferric ions, siderophores are categorized into three main groups: Catecholates (or phenolates such as enterobactin), hydroxymates (e.g., ferrichromes), and mixed type (e.g., pyoverdines). To date, roughly 500 siderophores have been identified, of which about 270 have been structurally characterized [1]. The majority of bacteria (both aerobic and facultative aerobic), fungi, and graminaceous plants excrete siderophores for iron scavenging [1] although it was proved that budding and fission yeasts do not produce such metabolites [2]. The primary role of siderophores is the mobilization, scavenging, and subsequent transport of iron into the cell. The indirect consequence of this strategy is that other, competing microorganisms are deprived of iron, hence, microorganisms able to produce siderophores gain an advantage in the environment. Since potentially toxic heavy metal ions (Ni, Cu, Co, Al, Cd, Ga, In, Pb, Zn, U, Np) stimulate siderophore production in many bacterial

species [3,4], it is believed that a secondary role of siderophores is associated with lowering the toxicity of these metals in the bacterial cell.

Nickel is one of the most produced base metals in the world—in 2019, the worldwide nickel production was 2.7 million tons, and the largest producers were Indonesia, Philippines and Russia [5]. Most of the produced nickel is used in stainless steel, but constantly more of it is also needed in battery production. Nickel is typically present in untreated industrial effluents [6], and it is highly toxic to aquatic organisms even at very low concentrations [7]. Nickel and other potentially toxic metals can be effectively removed from water, e.g., by ion exchange or reverse osmosis. However, these methods are expensive and consume high amounts of energy [6].

Peat soils are a potentially good source of microbes that possess efficient metal uptake mechanisms. Peat soils contain significant amounts of humic substances, capable of forming complexes with metal ions (including iron). Most of the iron in peatland is, therefore, in the form of chelates and only approximately 4–5% of the total iron is soluble and exchangeable [8]. The approximate content of free soluble ferrous ions decreases gradually from 200 mg g$^{-1}$ at a depth of 2.5 m to nearly zero in the upper layers of peat [9]; hence, efficient siderophore producers are most likely to be found near the peat surface. *Sphagnum* biomass is the main component of boreal peatlands, and it is therefore, an essential part of the boreal bioeconomy [10]. Peatlands are located in Alaska, northern Canada, northern Europe, and Siberia [11], covering a global area of about 10 million square kilometers [12].

Although peat soils have been extensively studied (i.e., vegetation, microbiota, chemical properties) there is a lack of information about siderophore production by boreal peat soil microbiota. The aim of the present research is to isolate bacteria able to produce siderophores efficiently from peat soil samples in a boreal climate (northern Finland). The employed isolation method is a modified variant of the method utilized by Nakouti et al. [13], where the main goal is to obtain Actinobacteria cultures. Actinobacteria are one of the dominant groups of microorganisms present in soil, and compared to other groups, are the most important producers of bioactive secondary metabolites [14]. In this study, different combinations of carbon and nitrogen sources are compared, in order to find the appropriate conditions for siderophore production by the isolates obtained.

The most promising isolate (in this case *Microbacterium* sp.) is cultivated in an optimal liquid medium, and the cells and siderophores obtained are immobilized on *Sphagnum* peat moss to develop a hybrid biosorbent for improved metal scavenging. Moss is known to have a lot of oxygen-containing functional groups, large specific area, and high porosity, all of which improve the ability of moss to bind contaminants via ion exchange and complexation [15,16]. In addition, microbial biomasses—either as viable or non-viable cells—can be utilized to uptake or bind metals [17]. This is the first study to evaluate the potential of a hybrid biosorbent containing three different metal-binding materials: *Sphagnum* moss, bacterial cells, and siderophores.

## 2. Materials and Methods

### 2.1. Isolation of Microorganisms from Peat Soil Samples with Emphasis on Actinobacteria

The soil samples were derived from 11 slices of soil (approximately 30 cm in length) provided by the Natural Resources Institute Finland (Table 1). All slices were collected in May 2016 in northern Finland (northern Ostrobothnia, Muhos, Ruostesuo, and Leppiniemi). Three samples from each slice were taken, each from a different depth (samples A: depth of 2–3 cm; samples B: depth of 13–15 cm; samples C: depth of 28–30 cm) measured from the top of the slice. The main goal of acquiring samples from different depths was to compare the siderophore production by microbiota-derived from layers differing in aeration level. In the case of soil strings nos. 1, 2, 3, 4, 7, 9, and 11, an orange-brown layer composed of ferric oxyhydroxides was clearly visible at a depth of 13–15 cm, indicating a low level of bioavailable iron. Approximately 1 g of soil was subsequently transferred to 10 mL of sterile 0.9% NaCl solution and the sample was shaken on a rotary shaker at room temperature (RT, 22 °C) for 2 h. The soil suspension was decanted to remove solid particles, and 1 mL of solution was transferred to an aseptic

Eppendorf-type tube. Next, the tube was incubated overnight at 40 °C, which is critical for selective isolation of Actinobacteria since heat exposure eliminates all of the non-sporulating bacteria present in the peat [13]. Subsequently, serial dilutions in 0.9% NaCl solution (100 µL of $10^{-5}$–$10^{-8}$) of the heat-treated sample were applied to Petri dishes containing Starch Casein Agar (SCA) medium (10.0 g soluble starch, 0.3 g casein, 2.0 g KNO$_3$, 0.05 g MgSO$_4$·7H$_2$O, 2.0 g K$_2$HPO$_4$, 2.0 g NaCl, 0.02 g CaCO$_3$, 0.01 g FeSO$_4$·7H$_2$O, 18.0 g agar L$^{-1}$ H$_2$O) supplemented with nystatin (50 µL mL$^{-1}$, final concentration) to prevent the growth of fungi. They were then incubated at 27 °C for up to four weeks. As a negative control, 100 µL of sterile, 0.9% NaCl solution was applied to SCA in a Petri dish. Most of the isolates (seven out of twelve) were obtained from the outer surface of the soil (depth A) located close to the moss roots, one isolate was from the middle depth (depth B), and four isolates were from the most deeply located piece of soil (depth C) (Table 1).

**Table 1.** Obtained isolates. Strains marked with brackets were further analyzed using 16S rDNA sequencing. A: depth of 2–3 cm, B: depth of 13–15 cm, and C: depth of 28–30 cm measured from the top of the soil slices.

| Isolate No. | Soil Slice No. | Depth | Visible Iron Precipitate in Soil | Dilution of the Sample |
|:---:|:---:|:---:|:---:|:---:|
| 1 | 11 | C | no | $10^{-7}$ |
| 2 | 11 | C | no | $10^{-7}$ |
| (3) | 5 | A | no | $10^{-8}$ |
| 4 | 7 | A | no | $10^{-5}$ |
| 5 | 3 | A | no | $10^{-6}$ |
| 6 | 3 | A | no | $10^{-6}$ |
| 7 | 3 | A | no | $10^{-6}$ |
| 8 | 3 | C | no | $10^{-6}$ |
| 9 | 4 | B | yes | $10^{-5}$ |
| 10 | 6 | A | no | $10^{-6}$ |
| (11) | 8 | C | no | $10^{-7}$ |
| (12) | 4 | A | no | $10^{-8}$ |

*2.2. Identification of the Isolates by 16S rDNA Sequencing*

Because of the similarity of the colony morphology of the isolated strains to the typical morphology of the actinobacterial colonies, three strains (3, 11, and 12) were selected for 16S rDNA sequencing analysis. Using the standard protocol [18], the genomic DNA of these strains was isolated from the cell pellets upon cultivation in a medium containing skimmed milk (lactose + casein) supplemented with asparagine. The sequencing analysis was performed at the Biocenter Oulu Sequencing Center. The 16S small subunit ribosomal gene was amplified with primers F519 (5-CAGCMGCCGCGGTAATWC-3) and R926 (5-CCGTCAATTCCTTTRAGTTT-3). The F519 primer contained an Ion Torrent adapter sequence A, a 9-bp unique barcode sequence, and one nucleotide linker. The R926 primer contained an Ion Torrent adapter trP1 sequence. Polymerase chain reaction (PCR) analyses were performed in 25 µL reactions in two replicates, which contained 1 × Phusion GC master mix (Thermo Scientific, Espoo, Finland), 0.4 µM of forward and reverse primers, and 20 ng of genomic DNA as the template. After an initial denaturation (3 min at 98 °C), the following cycling conditions were used: twenty-eight cycles of 98 °C, 10 s; 64 °C, 20 s; 72 °C, 20 s. After PCR, the samples were purified with AMPure XP reagent (Agencourt Bioscience, Brea, CA, USA). The amplicon concentration of the pure samples was measured on a Bioanalyzer DNA-1000 chip (Agilent Technologies, Santa Clara, CA, USA) and the samples were pooled in equivalent amounts. The pooled samples were further purified with Ampure XP, and sequencing was performed with Ion Torrent PGM on a 316 chip using Ion View chemistry (ThermoFisher Scientific, Waltham, MA, USA).

## 2.3. Isolation of Pure Strains from Mixed Cultures

The 16S rDNA sequencing data showed that all strains selected for identification constituted mixed cultures of two different bacterial species: *Bacillus* sp. and *Microbacterium* sp. Therefore, the strains were isolated by serial dilution and the streak plate method on Lysogeny broth (LB) agar (10.0 g tryptone, 5.0 g yeast extract, 10.0 g NaCl, 15.0 g agar L$^{-1}$ H$_2$O) plates. When different kinds of colonies were detected, the obtained strains were re-streaked on the Petri dishes containing LB agar from which the samples for liquid media inoculation were taken with a sterile, disposable loop.

Both *Bacillus* and *Microbacterium* are gram-positive rods and the clearest difference between their growth is that *Bacillus* spp. are spore-forming and *Microbacterium* spp. are non-spore-forming bacteria. Thus, after both strains were cultivated overnight in liquid LB medium, the liquid cultures were incubated for 10 min at 80 °C, which eliminates all non-sporulating bacteria, in this case, the *Microbacterium* sp. Afterwards, the cultures were plated again on LB agar plates and incubated overnight to detect the growth of the sporulating strains.

## 2.4. Cultivation of the Obtained Isolates in Media Containing Various Carbon and Nitrogen Sources

Ten liquid media containing different combinations of carbon and nitrogen sources (glucose, fructose, asparagine, glutamic acid, glycerol, succinic acid, urea, ammonium sulfate, and skimmed milk) were compared to find appropriate conditions for siderophore production by the obtained isolates. The compositions of the media are presented in Table 2. All of the media were deprived of iron, which is the most important precondition for efficient siderophore production in synthetic media [13]. The obtained isolates were cultivated in 20 mL of the given medium in conical flasks placed on a rotary shaker with moderate agitation (Infors Multitron, Infors HT), at RT (22 °C). As temperature is one of the most important factors among those that are crucial for efficient synthesis of siderophores [19], and due to the fact that the obtained strains were isolated from soil sampled in a cold climate area, incubation was carried out at RT for seven days. All of the media components were obtained from Sigma-Aldrich (St. Louis, MO, USA) and were of molecular biology grade.

**Table 2.** Composition of the media investigated for siderophore production.

| Ingredient [g L$^{-1}$] | G-ASN | G-GLU | GLY-ASN | GLY-GLU | M-ASN | M-GLU | F-ASN | F-GLU | G-U | SA |
|---|---|---|---|---|---|---|---|---|---|---|
| L-Asn (mono-hydrate) | 2.0 | - | 2.0 | - | 2.0 | - | 2.0 | - | - | - |
| L-Glu | - | 2.0 | - | 2.0 | - | 2.0 | - | 2.0 | 1.0 | - |
| Urea | - | - | - | - | - | - | - | - | 0.85 | - |
| (NH$_4$)$_2$SO$_4$ | - | - | - | - | - | - | - | - | - | 1.0 |
| Glycerol (100%) [mL] | - | - | 5.55 | 5.55 | - | - | - | - | - | - |
| Glucose | 7.0 | 7.0 | - | - | - | - | - | - | 10.0 | - |
| Skimmed milk | - | - | - | - | 7.0 | 7.0 | - | - | - | - |
| Fructose | - | - | - | - | - | - | 7.0 | 7.0 | - | - |
| Succinic acid | - | - | - | - | - | - | - | - | - | 4.0 |
| K$_2$HPO$_4$ | - | - | - | - | - | - | - | - | 0.56 | 6.0 |
| Na$_2$HPO$_4$ | 0.96 | 0.96 | 0.96 | 0.96 | 0.96 | 0.96 | 0.96 | 0.96 | - | - |
| KH$_2$PO$_4$ | 0.44 | 0.44 | 0.44 | 0.44 | 0.44 | 0.44 | 0.44 | 0.44 | - | 3.0 |
| MgSO$_4$·7H$_2$O | 0.2 | 0.2 | 0.2 | 0.2 | 0.2 | 0.2 | 0.2 | 0.2 | 0.2 | 0.2 |

G-ASN—glucose-asparagine, G-GLU—glucose-glutamic acid, GLY-ASN—glycerol-asparagine, GLY-GLU—glycerol-glutamic acid, M-ASN—skimmed milk-asparagine, M-GLU—skimmed milk-glutamic acid, F-ASN—fructose-asparagine, F-GLU—fructose-glutamic acid, G-U—glucose-urea, SA—succinic acid-ammonium sulfate, L-Asn—L-asparagine, L-Glu—L-glutamic acid.

## 2.5. Siderophore Production

### 2.5.1. Preparation of CAS-Fe Reagents

Evaluation of siderophore production efficiency was based on the Chrome Azurol S (CAS) method initially developed by Schwyn and Neilands [20], and its variants [21–23]. Briefly, the way that CAS assay works is based on the removal of ferric ions from the CAS-Fe complex by a strong chelator

(such as a siderophore), which results in a color change from blue/green to yellow. The CAS-Fe reagents were prepared by mixing the solutions mentioned in Table 3 in the following order: D + C + B (+ A). The order of mixing is of crucial importance as it ensures the formation of appropriate complexes. Next, 3 g of agar (Bacto™Agar, BD Cat. No. 214010, USA) was added to obtain the solid CAS reagent, and the pH of the solution was adjusted to 7.0 with NaOH. The CAS-Fe suspensions obtained were sterilized (121 °C, for 20 min) and the agar suspension was portioned out into the Eppendorf-type tubes (1 mL per tube). Shortly before assessment of the siderophore content, holes were made in the solid CAS-Fe agar to ensure the fast saturation of the dye with the spent medium.

**Table 3.** Composition of the solid and liquid CAS reagents.

| Ingredient | Solid | Liquid |
|---|---|---|
| **Solution A** | | |
| MOPS | 2.1 g | - |
| ddH$_2$O | 70 mL | - |
| **Solution B** | | |
| HDTMA (CTAB) | 7.3 mg | 72.9 mg |
| 10 mM HCl solution | 10 mL | 40 mL |
| **Solution C** | | |
| FeCl$_3$·6 H$_2$O | 0.27 mg | 0.27 mg |
| 10 mM HCl solution (200 μL 0.5N HCl + 9.8 ml ddH$_2$O) | 10 mL | 10 mL |
| **Solution D** | | |
| CAS | 5.18 mg | 60.5 mg |
| ddH$_2$O | 10 mL | 50 mL |
| **Agar** | 3 g | - |

CTAB—cetrimonium bromide, HDTMA—hexadecyltrimethylammonium bromide, MOPS—3-(N-morpholino) propanesulfonic acid.

### 2.5.2. CAS Assay with Spent Media

After seven days of cultivation of the isolates in the media described in Table 2, 1 mL of each suspension was centrifuged (10,000 rcf, 5 min). The pellet was discarded, and 100 μL of the supernatant was added into the top of the CAS-Fe agar tubes. The blue to yellow discoloration level, indicating the presence of siderophores in the spent media was observed for approximately 48 h. The evaluation of discoloration levels was based on subjective observation with the naked eye, and the results were ascribed to the four-level color change scale, (Figure 1): +++ – high level of discoloration, ++ – medium level, + – low level, and (+) – scarcely visible level of discoloration. Additionally, a negative control was performed with the corresponding uninoculated medium incubated under the same conditions as the inoculated medium (shown as (−)).

Liquid CAS assay is based on spectrophotometry [22]. The method was used before and after the immobilization of siderophores on *Sphagnum* moss to determine the number of immobilized siderophores. Liquid CAS-Fe solution (100 μL) and cell-free supernatant (100 μL) were pipetted onto a 96 microwell plate. The results were read after 20 min at a wavelength of 620 nm using a plate reader (Victor3, PerkinElmer). The number of siderophores produced was measured in percent siderophore units (psu), calculated according to the following Equation (1) from three parallel absorbance measurements:

$$Siderophore\ production\ (psu) = \frac{(A_r - A_s) \times 100}{A_r} \tag{1}$$

where $A_r$ is the absorbance of the reference (CAS solution and uninoculated medium) and $A_s$ is the absorbance of the sample (CAS solution and cell-free supernatant of sample) [22,24].

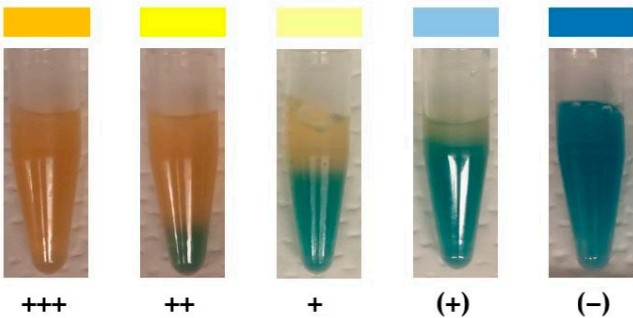

**Figure 1.** Blue to yellow discoloration levels shown in decreasing intensity from left to right. +++ –high level of discoloration, ++ –medium level, + –low level, (+)–scarcely visible level of discoloration and (−)—negative control for the corresponding, uninoculated medium.

### 2.6. Immobilization on Sphagnum Peat Moss

The *Microbacterium* sp. cells and/or siderophores cultivated in G-GLU medium were immobilized on *Sphagnum* peat moss in conical flasks placed in a rotary shaker with an agitation of 120 rpm at RT for 24 h. For the immobilization of siderophores, bacterial cells were removed by centrifugation (11,000 rcf, 5 min, 4 °C). The moss (0.25 g) and 50 mL of the bacterial solution or bacteria-free supernatant were added to the conical flasks (moss concentration 5 g L$^{-1}$). Three parallel immobilization tests and a reference experiment with 0.9% NaCl solution were performed. After the immobilization, the hybrid biosorbents produced were filtered out of the solutions with a 150 µm wire mesh and further utilized in nickel biosorption. To evaluate the efficiency of the immobilization and to avoid any interference caused by the biosorbent particles, the filtrates (containing unimmobilized cells and siderophores) were sampled for CAS analysis, optical density (OD) measurement, cell dry weight analysis, and plating. The immobilization percentage $Y_i$ was calculated according to Equation (2), where $X_i$ is the concentration of immobilized cells (mg L$^{-1}$), and $X_t$ is the total concentration of cells [25]:

$$Y_i = \frac{X_i}{X_t} \times 100\%$$ (2)

### 2.7. Nickel Biosorption

The hybrid biosorbents were tested in the sorption of nickel from solution with a Ni concentration of 18 mg L$^{-1}$ at pH 6. The Ni solution was prepared by dissolving nickel nitrate (Ni(NO$_3$)$_2 \cdot$ 6 H$_2$O) in ultrapure Milli-Q water. The solution was sterilized by autoclaving (121 °C, 20 min) before the biosorption experiments.

The hybrid biosorbents were added into 45 mL of the Ni solution (biosorbent concentration approx. 5.56 g L$^{-1}$) and were shaken 20 rpm for 72 h at RT. Three parallel sorption experiments were conducted. After biosorption, the solutions were centrifuged (700 rcf, 10 min) and 10 mL of the supernatant, with 50 µL of 65% HNO$_3$ added, was used for nickel analysis. Nickel concentrations were measured by inductively coupled plasma optical emission spectrometry (ICP-OES) according to the standard method [26].

The sorption capacities $q$ (mg g$^{-1}$) were calculated according to Equation (3), where $C_0$ is the initial concentration of metal ions (mg L$^{-1}$), $C_r$ is the residual concentration of metal ions (mg L$^{-1}$), $V$ is the volume of the solution (L), and m is the mass of the biosorbent (g) [27]:

$$q = \frac{(C_0 - C_r)V}{m}$$ (3)

## 3. Results

### 3.1. Identification of the Obtained Isolates

Twelve isolates were obtained on the Petri dishes containing SCA broth supplemented with nystatin (Table 1). Soil samples from slices 1, 2, 9, and 10 produced weak growth on the SCA Petri dishes. A large diversity of growth morphology was observed between the obtained isolates. As the primary goal of the research was to isolate Actinobacteria, isolates 3, 11, and 12 were selected for 16S rDNA sequencing analysis because of the similarity of their colony morphology (cream-colored filamentous growth) to that of Actinobacteria. The morphology of all isolates is described in detail in the Supplementary Materials.

Although high levels of serial dilution were applied prior to inoculation of the Petri dishes, the 16S rDNA sequencing data showed that all of the strains selected for identification constituted mixed cultures. Strain 3 was a mixed culture of *Bacillus* sp. and *Microbacterium* sp., and strains 11 and 12 were a mixed culture of *Bacillus cereus* and *Microbacterium* sp. All of the pure isolated strains showed different growth patterns, which are described in the Supplementary Materials. *Microbacterium* sp. from mixed culture 11 had yellow colonies with a wrinkled surface and a brown halo around them. In liquid cultivations, isolate 11 *Microbacterium* formed flocs.

### 3.2. Evaluation of Siderophore Production and Immobilization Efficiency

The efficiency of siderophore production and excretion by microorganisms strongly depends on factors, such as low iron stress, pH, temperature, the nature of the carbon and nitrogen sources, availability of phosphorus, and oxygen transfer [19,28]. Although the purpose of this study was to isolate Actinobacterial species, all the isolates were tested for siderophore production, since *Bacillus* spp. are also known to form various types of siderophores [29,30]. To select the most efficient siderophore producers, the obtained isolates were cultivated in liquid media deprived of iron, in the presence of various carbon and nitrogen sources (Table 2).

Table 4 presents the results of the CAS assay performed with the spent media obtained immediately after isolate cultivation. The best siderophore production results were obtained with isolates from the most deeply located piece of soil (28–30 cm, depth C) where the aeration level would have been the lowest. Also, most of the isolates obtained from the outer surface of the soil (2–3 cm, depth A) produced a high level of discoloration in the CAS assay, but there was more variation in the results. The only isolate obtained from an orange-brown layer composed of ferric oxyhydroxides was isolate 9, which produced a large number of siderophores only in the media containing skimmed milk supplemented with either asparagine or glutamic acid (M-ASN and M-GLU, respectively). It is worth stressing that a medium appropriate for microorganism growth is not always the one in which siderophore production efficiency is the highest. The *Microbacterium* sp. from mixed culture 11 was able to produce siderophores in all of the tested media, although in the medium containing succinic acid as a sole carbon source in combination with ammonium sulfate (SA medium, Tables 2 and 4) only a scarcely visible indication of siderophore excretion was observed. With other isolates no indication of siderophore production was observed in the SA medium. Succinic acid has been proven to stimulate siderophore biosynthesis in *Pseudomonas aeruginosa* as it influences metabolic reactions leading to pyoverdine production [19].

It can be inferred from Table 3 that almost all of the mixed and pure cultures produced siderophores in the M-ASN and M-GLU media. The *Microbacterium* sp. isolated from mixed culture 11 produced siderophores efficiently in the media containing fructose or glycerol supplemented with glutamic acid (F-GLU and GLY-GLU), but only slightly visible discoloration was observed when cultivated in the media supplemented with asparagine (F-ASN and GLY-ASN). Also, isolate 1 produced siderophores efficiently in the F-GLU medium and only slightly visible discoloration when cultivated in the F-ASN medium. Promising results were obtained in the case of media containing glucose supplemented with either asparagine or glutamic acid (G-ASN and G-GLU, respectively). All of the mixed cultures

excreted siderophores in these media, and the *Microbacterium* sp. from isolate 11 formed high levels of siderophores. Spent medium obtained after cultivation of isolate 9 in the G-GLU medium did not produce CAS-Fe agar discoloration, and similar behavior was observed in the instance of isolate 12 cultivated in the G-ASN medium. Only one of the pure cultures (isolate 11 *Microbacterium*) produced high levels of siderophores in each of the media. *B. cereus* from isolate 11 did not produce siderophores at all, and other pure strains produced only a scarcely visible level of CAS-Fe agar discoloration when cultivated in the media containing skimmed milk.

**Table 4.** Discoloration levels observed upon addition of spent media to test tubes containing CAS-Fe agar. (−) indicates no visible discoloration, (+) depicts a scarcely visible level of discoloration, + low level, ++ medium level and +++ high level of discoloration. The letters A, B, and C next to the isolate number indicate the depth of sampling: 2–3 cm, 13–15 cm, and 28–30 cm, respectively. The letters B. and M. after the depth of sampling indicate the genera *Bacillus* and *Microbacterium*.

| Isolate No. | G-ASN | G-GLU | GLY-ASN | GLY-GLU | M-ASN | M-GLU | F-ASN | F-GLU | G-U | SA |
|---|---|---|---|---|---|---|---|---|---|---|
| 1C | +++ | +++ | (+) | ++ | ++ | ++ | (+) | +++ | + | − |
| 2C | +++ | +++ | + | ++ | +++ | +++ | − | ++ | (+) | − |
| 3A | (+) | + | − | + | +++ | +++ | (+) | + | + | − |
| 3A B. | − | − | − | − | (+) | − | − | − | − | − |
| 3A M. | − | − | − | − | (+) | (+) | − | − | − | − |
| 4A | +++ | +++ | ++ | +++ | +++ | +++ | ++ | ++ | (+) | − |
| 5A | + | + | +++ | +++ | +++ | +++ | +++ | +++ | + | − |
| 6A | + | + | (+) | ++ | +++ | +++ | + | + | (+) | − |
| 7A | +++ | +++ | +++ | +++ | +++ | +++ | +++ | +++ | ++ | − |
| 8C | +++ | +++ | +++ | +++ | +++ | +++ | +++ | +++ | + | − |
| 9B | + | − | − | − | +++ | +++ | (+) | + | − | − |
| 10A | + | +++ | + | ++ | +++ | +++ | + | ++ | + | − |
| 11C | ++ | +++ | +++ | ++ | ++ | ++ | +++ | ++ | ++ | − |
| 11C B. | − | − | − | − | − | − | − | − | − | − |
| 11C M. | +++ | +++ | (+) | +++ | (+) | + | (+) | +++ | (+) | (+) |
| 12A | − | ++ | − | ++ | ++ | ++ | + | ++ | + | − |
| 12A B. | − | − | − | − | (+) | (+) | − | − | − | − |
| 12A M. | − | − | − | − | (+) | (+) | − | − | − | − |

The most promising results were obtained in the case of the *Microbacterium* sp. from isolate 11. Formation of the brown pigment by the *Microbacterium* sp. was highest in the same media that supported siderophore production. The only clear difference was in the case of the medium supplemented with glucose and urea (G-U). When the *Microbacterium* sp. was cultivated in the G-U medium, it produced only a scarcely visible level of discoloration in the CAS assay, but the amount of brown pigment was as high as in the best media for siderophore production.

According to the liquid CAS assay, the number of siderophores produced by *Microbacterium* sp. in the G-GLU medium was 73 psu (Figure 2). The immobilization of siderophores on *Sphagnum* peat moss was efficient–the average immobilization rate was 85% when only siderophores were immobilized (248 psu g$^{-1}$ of moss) and 92% when both cells and siderophores were immobilized (268 psu g$^{-1}$ of moss). However, according to the *t*-test, the difference between immobilization efficiencies is not statistically significant (*p*-value > 0.05). Based on the cell dry weight measurements, it could be detected that the number of cells decreased during immobilization (Figure 2), which indicates the successful attachment of cells on the moss. The average bacterial cell immobilization percentage of three parallel immobilization tests was 48%.

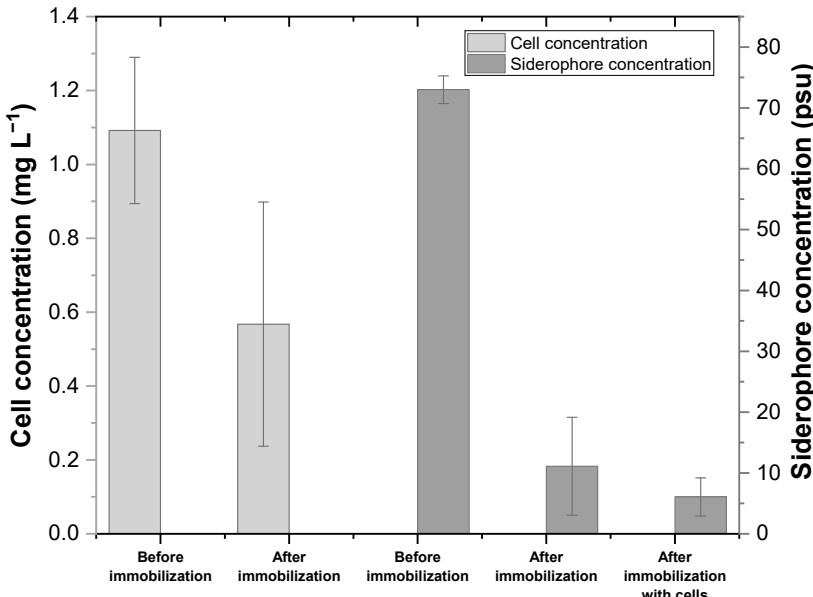

**Figure 2.** Immobilization of siderophores and cells of isolate 11 *Microbacterium* on *Sphagnum* moss. Cell dry weight concentration (mg mL$^{-1}$) and siderophores (psu) in cultivation media (G-GLU) before and after the immobilization experiments are presented. Each series contains the average value of three parallel immobilization experiments with standard deviations.

*3.3. Evaluation of the Nickel Binding Efficiency of Hybrid Biosorbents*

The nickel binding efficiency of hybrid biosorbents that contain cells and/or siderophores from isolate 11 *Microbacterium* immobilized on *Sphagnum* moss was tested at pH 6 with a Ni concentration of 18 mg L$^{-1}$ (Figure 3). The highest capacity, 3.11 ± 0.01 mg g$^{-1}$, was achieved with the biosorbent containing only moss and siderophores (Ni removal 96.0%). The capacities of the biosorbent containing both cells and siderophores and raw moss were 3.08 ± 0.02 mg g$^{-1}$ and 2.99 ± 0.01 mg g$^{-1}$ (Ni removal 94.9% and 92.4%), respectively. According to the *t*-test, the differences between Ni removal levels achieved with both hybrid biosorbents and raw moss are statistically significant (*p*-value < 0.05).

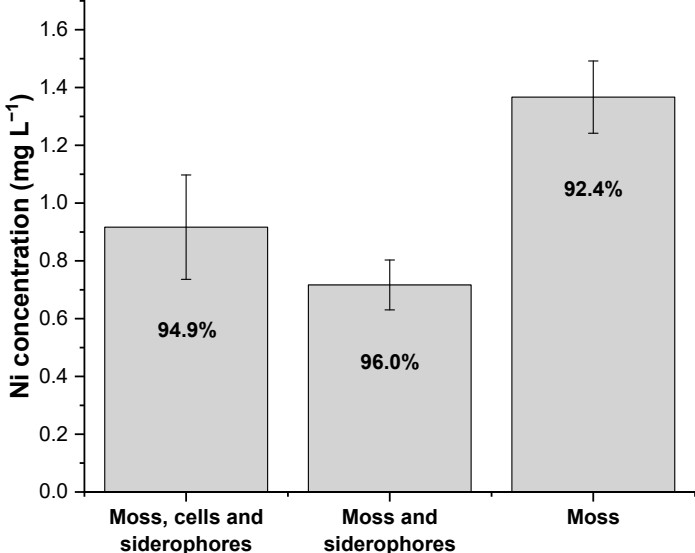

**Figure 3.** Achieved residual Ni concentrations (columns) and removal rates (%) with the biosorbents. Cells and siderophores of isolate 11 *Microbacterium* were used in the hybrid biosorbents. The initial Ni concentration was 18 mg L$^{-1}$. Results of three parallel biosorption tests with standard deviations are presented.

## 4. Discussion

Most soil bacteria belong to four main phylogenetic groups: Proteobacteria, Firmicutes, Actinobacteria, and Bacteroidetes. Although the relative content of various bacterial genera is strongly dependent on the origin of the soil and on the climate, it can be concluded that the aforementioned microorganisms are almost always present in the soil of any type [31].

In the present research, a method aimed at the isolation of Actinobacteria was applied to peat soil samples. The thermal pretreatment of the samples prior to Petri dish inoculation was to eliminate the non-sporulating bacteria present in the soil. Among the analyzed strains, three mixed cultures consisting of *Bacillus* sp. (including *Bacillus cereus*) with *Microbacterium* sp. were identified. As mentioned before, within these two genera *Bacillus* is spore-forming and *Microbacterium* non-spore-forming. Thus, the temperature of 40 °C was too low to eliminate *Microbacterium* spp., since the 80 °C temperature used afterwards to differentiate the strains eliminated the *Microbacterium* spp. Considering the average qualitative content of soil bacteria belonging to different genera in an average soil sample, obtaining cultures containing *Bacillus* sp. and *Microbacterium* sp. is unsurprising.

Of the identified species, only *Microbacterium* sp. belongs to Actinobacteria. The isolate 11 *Microbacterium* secreted brown pigmentation around its colonies and in all of the tested liquid media. Bacteria can produce different kinds of dark pigments [32]. The most commonly produced pigments are various types of melanins, which are humic-type compounds that can be involved in the metal uptake of microorganisms [33]. They include allomelanins, eumelanins, and pheomelanins. One of several allomelanins is pyomelanin, which derives from homogentisic acid (HGA) as a result of autooxidation and self-polymerization. Another melanin pigment commonly secreted by microorganisms is dihydroxyphenylalanine (DOPA) melanin. [34] According to Turick et al., the coloration of pyomelanin pigments is reddish-brown, and they are readily diffusive through agar media, whereas DOPA melanin pigments are dark or black and non-diffusive [33]. The coloration and diffusion of the pigment produced by the isolated *Microbacterium* sp. indicates that the pigment could possibly be pyomelanin. However, since the resolution of 16S rDNA analysis did not allow us to determine the species of the *Microbacterium* isolated, it can only be speculated that the strain in question expressly secretes pyomelanin. The pigment formation of *Microbacterium* sp. was obviously related to the iron uptake, since the darker the spent medium, the higher the discoloration of the CAS-Fe agar. On the other hand, pigment production was also high in the medium containing glucose supplemented with urea (G-U), but in this instance discoloration of the CAS-Fe agar remained low. This indicates that glucose increased both siderophore and pigment production but urea as a nitrogen source supported only the production of the brown pigment. Zheng et al. reported that pyomelanin secretion of *Legionella pneumophila* was inversely related to siderophore activity, since pyomelanin mediates the reduction of ferric iron to ferrous iron, which is soluble, and hence, the bioavailable form of iron [35].

The only pure strain that was clearly producing siderophores in the present research was the *Microbacterium* sp. isolated from mixed culture 11. Even though Actinobacteria are widely known to produce siderophores and other important metabolites, there is a lack of information about the types of siderophores produced by *Microbacteria*. The majority of the studies concerning *Microbacterium* spp. and siderophores deal with siderophore auxotrophic bacteria, which means that the bacteria are only able to take up siderophores produced by other microorganisms and not to form their own siderophores [36–38]. However, some siderophore-producing *Microbacterium* strains have been detected, such as *M. panaciterrae* [39]., *M. liquefaciens* which was observed to increase Fe solubilization in serpentine soil [40], and *M. oleivorans* which has been even hypothesized to bind uranium with its siderophores [41]. In addition, *M. oxydans* and *M. liquefaciens* have been detected to have the capability of removing nickel and vanadium in a liquid culture [42].

Supplementation of amino acids generally stimulates siderophore biosynthesis, although their influence may vary from strain to strain. This was noticeable in the instance of the *Microbacterium* sp. isolated from mixed culture 11. The highest siderophore production (measured by CAS-Fe

discoloration level) by isolate 11 *Microbacterium* was observed in the instance of media containing glucose supplemented with asparagine or glutamic acid (G-ASN and G-GLU), and those containing glycerol or fructose supplemented with glutamic acid (GLY-GLU and F-GLU). In the G-GLU medium, the number of siderophores produced was 73 psu, whereas Arora and Verma reported that the siderophore production values of various bacteria isolated from soil varied from 28 to 70 psu [22]. It can be inferred from the results of siderophore production efficiency in different media that the nature of both the nitrogen and carbon source is critical for efficient siderophore production. Moreover, it is commonly known that the efficiency of siderophore production strongly depends on factors, such as the concentration of bioavailable iron, pH, and temperature. For example, the properties of metal ions and functional groups of biosorbent can be affected by the pH [43]. The discrepancies between the results of experiments with *Pseudomonas* species clearly show that even the same species isolated from different environments or derived from different culture collections may differ in siderophore production efficiency. Prabhu and Bindu reported that, in the case of *Pseudomonas,* the highest siderophore production efficiency was observed in the medium containing glycerol and ammonium chloride [44]. Whereas, Duffy and Defago reported that glycerol did not lead to a significant increase in pyochelin production by *P. fluorescens* [45].

Similarly, the incubation temperature is a critical factor influencing the efficiency of siderophore production, although it may differ substantially even for the same species (but of different origin). It was shown that *Pseudomonas putida* isolated from tropical areas grew normally and produced siderophores at 35 °C, whereas the same species isolated from a temperate climate area showed neither intensive growth nor siderophore production [45,46]. However, among the identified strains described herein, none belonged to *Pseudomonas*, the above-mentioned remarks can be associated with general, highly diverse tendencies of microorganisms to siderophore production. All of the present mixed cultures produced siderophores in the media containing skimmed milk (casein + lactose) in combination with either asparagine (M-ASN) or glutamic acid (M-GLU). The most probable explanation for this result is the fact that, apart from sugars, skimmed milk contains other nutrients, such as salts which influence the growth of the cells, and hence, indirectly affect siderophore metabolism. When considering the general influence of a medium composition containing glycerol on the siderophore production by all strains, glycerol most positively affected siderophore efficiency in combination with glutamic acid (GLY-GLU), whereas in the presence of asparagine its influence was weaker (GLY-ASN) (Table 3).

The immobilization of the *Microbacterium* sp. siderophores on moss was slightly more efficient in the presence of the bacterial cells. With low iron availability, siderophores are usually excreted outside the cell to form complexes with ferric iron, which are then transported to cytosol [47]. In this case, the siderophores might have moved inside the cells in the same way, since the exact composition of the moss utilized is unknown, and it might have contained metals with which the siderophores might have formed complexes. Furthermore, the nickel biosorption was slightly more efficient when only the siderophores were immobilized (Figure 3). Even though microbial biomasses are known to have the potential for binding contaminants [17], in this study the use of microbial cells in hybrid biosorbents did not increase the sorption efficiency compared to the use of siderophores alone. It can be presumed that the presence of the cells only improved the attachment of siderophores on the moss surface.

The sorption capacity of the biosorbent increases when the metal concentration in the solution increases [48,49], or the biosorbent concentration decreases [50]. These effects, shown in Table 5, where some capacities obtained with different biosorbents are listed. The hybrid biosorbents developed provided high removal rates for nickel, but the capacities remained low. However, one must bear in mind that the capacities obtained in this study are not the maximum capacities reported for some biosorbents.

**Table 5.** Nickel sorption capacities and removal rates achieved with different biosorbents under different conditions.

| Biosorbent | Concentration of Biosorbent (g L$^{-1}$) | $C_0$ (mg L$^{-1}$) | pH | T (°C) | q (mg g$^{-1}$) | Ni Removal (%) | Reference |
|---|---|---|---|---|---|---|---|
| *Sphagnum* peat moss and *Microbacterium* sp. cells and siderophores | 5.56 | 18 | 6 | 22 | 3.08 ± 0.02 | 94.9 | This study |
| *Sphagnum* peat moss and *Microbacterium* sp. siderophores | 5.56 | 18 | 6 | 22 | 3.11 ± 0.01 | 96.0 | This study |
| *Sphagnum* peat moss | 5.56 | 18 | 6 | 22 | 2.99 ± 0.01 | 92.4 | This study |
| *Streptomyces rimosus* cells | 3 | 20 | 5.7 | 20 | 6 | 90.0 | [49] |
| *Streptomyces rimosus* cells | 3 | 100 | 5–5.7 | 20 | 16.3 | 48.9 | [49] |
| *Bacillus thuringiensis* cells | 1.0 | 119 | 6 | 35 | 21.5 | 18.0 | [48] |
| *Arthrobacter* sp. cells | 2.8 | 125 | 5–5.5 | 30 | 10.2 * | 22.8 | [51] |
| *Arthrobacter* sp. cells | 1.4 | 35 | 5–5.5 | 30 | 9 | 36.0 | [51] |
| *Arthrobacter* sp. cells | 1.4 | 150 | 5–5.5 | 30 | 12.7 * | 11.9 | [51] |
| *Streptomyces coelicolor* cells | 1 | 148 | 8 | 25 | 11.1 | 7.5 | [52] |
| Peat | 1 | 20 | 5.6-6 | 20 | 10 | 50.0 | [27] |
| Peat | 1 | 60 | 5.6–6 | 20 | 16 * | 26.7 | [27] |
| Peat | 4 | 200 | 7 | 25 | 9.18 * | 15.2 | [50] |

$C_0$—initial nickel concentration, q—sorption capacity, *—maximum sorption capacity.

## 5. Conclusions

The general conclusion of this study, corroborating many other previous research outcomes described in the literature, is the necessity for optimizing media composition intended for efficient siderophore production, even in the case of the same species derived from different sources. The most appropriate media compositions for siderophore production by the obtained *Microbacterium* sp. on laboratory scale were found. The pure culture of *Microbacterium* sp. was further studied in nickel biosorption experiments using immobilized cells and siderophores. Sphagnum peat moss proved to be a well suitable carrier for *Microbacterium*; since, on average, 48% of the cells and 92% of the siderophores produced by the *Microbacterium* sp. attached to the moss. The hybrid biosorbents developed were found to be efficient for nickel scavenging from water, and could thus, be further studied in the bioremediation of real metal-rich effluents as a final polishing step before discharge to the environment. In addition, since pH is considered one of the most important factors to affect biosorption, its effect should be investigated further.

**Supplementary Materials:** The following are available online at http://www.mdpi.com/2073-4441/12/7/2000/s1, Figure S1: Colony morphology of the isolates obtained; Figure S2: Colony morphology of the pure strains.

**Author Contributions:** Conceptualization, H.V., M.B., S.T., T.L., M.H. and V.-H.S.; validation, H.V., M.B. and M.H.; formal analysis, H.V.; investigation, H.V., M.B. and M.H.; writing—original draft preparation, H.V. and M.B.; writing—review and editing, S.T., T.L., M.H., V.-H.S. and J.T.; visualization, H.V. and M.B.; supervision, S.T. and T.L.; project administration, S.T.; funding acquisition, S.T. All authors have read and agreed to the published version of the manuscript.

**Funding:** This research was funded by the European Regional Development Fund, project A71699 "Sustainable refining of peatland biomasses to valuable products". The study was also part of the "Supporting Environmental, Economic and Social Impacts of Mining Activity" (KO1030 SEESIMA) research project and received financial support from the Kolarctic CBC (Cross-Border Collaboration), the European Union, Russia, Norway, Finland, and Sweden. Its contents are the sole responsibility of the authors at the University of Oulu, and do not necessarily reflect the views of the European Union or the participating countries. The sequencing costs were covered by the financing of Maa- ja vesitekniikan tuki (MVTT) in the project entitled "Microbes in constructed forest wetlands as biocatalysts in wastewater treatment".

**Acknowledgments:** Juha Piispanen at the Natural Resource Institute Finland is acknowledged for the peat samples. The authors also wish to thank the Biocenter Oulu Sequencing Center for the 16S rDNA sequencing services.

**Conflicts of Interest:** The authors declare no conflict of interest. The funders had no role in the design of the study; in the collection, analyses, or interpretation of data; in the writing of the manuscript, or in the decision to publish the results.

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
