# Peer review of "Isolation of Efficient Metal-Binding Bacteria from Boreal Peat Soils and Development of Microbial Biosorbents for Improved Nickel Scavenging"

_water, doi:10.3390/w12072000_

Round 1
Reviewer 1 Report
The authors report the isolation of metal-binding bacteria from boreal peat soil. The introduction of the study is well written for the background and necessity of the study. But it could be improved if the authors explain how the nickel in water become the problems and its situation. The methodology is sound. The results are presented well and discussed appropriately. However, it needs to improve for the readers. For Fig. 2 why the authors show the colony picture that does not show any particular points. If they wan to show that, Fig. 2 needs more information.
Author Response
Thank you for the comments. Information about the problems and current situation of nickel production is added to the introduction part (lines 46–52). In addition, Fig. 2 from line 245 is moved to the supplementary material.
Reviewer 2 Report
The present manuscript studies the isolation of bacteria with the capacity to produce siderophores from a peat soil. In general, the manuscript is correct, well organized and the results are consistent.
Comments:
1) The authors indicate their preference for the group of actinomycetes, however they also consider Bacillus as a result, this is a bit contradictory. From my point of view, that restriction should be removed in the manuscript.
2) The most criticizable part is that related to immobilization. I think that it should not be considered as a true immobilization but as a mixture of biosorbents. What would happen if the immobilization efficiency was measured after shaking the mixture instead of measuring a filtrate? Authors should consider this problem, the results would not be affected.
3) Unify the centrifugation units. Better to express these units in g or rcf.
4) In biosorption experiments, it is necessary to indicate the shaking speed.
Author Response
1) Thank you for pointing this out. The most interesting strain isolated in the study was Microbacterium sp., which belongs to Actinobacteria, and this strain was further chosen for biosorption tests. Some Bacillus spp. were also found, and they were tested for siderophore production since Bacillus strains are known to form siderophores (addition in the manuscript on lines 250–252). However, our purpose was not to focus on Bacillus spp., since the results from CAS assay were not promising. Some parts discussing Bacillus spp. are now removed (lines 237-244 and 361-368).
2) Thank you for the suggestion. The immobilization efficiency was measured using the filtrate to avoid any interference caused by the moss (biosorbent particles). Unimmobilized siderophores and cells can pass through the 150 µm wire mesh, but binded siderophores and cells cannot. This is now defined more closely in the manuscript on lines 202–203.
3) The centrifugation unit on line 214 is now expressed as rcf.
4) The shaking speed is added to the manuscript on line 213.